# The Impact of Serum Protein Adsorption on PEGylated NT3–BDNF Nanoparticles—Distribution, Protein Release, and Cytotoxicity in a Human Retinal Pigmented Epithelial Cell Model

**DOI:** 10.3390/pharmaceutics15092236

**Published:** 2023-08-30

**Authors:** Maria Dąbkowska, Alicja Kosiorowska, Bogusław Machaliński

**Affiliations:** 1Independent Laboratory of Pharmacokinetic and Clinical Pharmacy, Rybacka 1, 70-204 Szczecin, Poland; alicjakosiorowska@gmail.com; 2Department of General Pathology, Pomeranian Medical University, Rybacka 1, 70-204 Szczecin, Poland; boguslaw.machalinski@pum.edu.pl

**Keywords:** brain-derived neurotrophic factor (BDNF), neurotrophin 3 (NT3), polyethylene glycol (PEG), biodegradable nanoparticles, ocular neurodegeneration, protein corona, human retinal pigmented epithelial cells (ARPE-19)

## Abstract

The adsorption of biomolecules on nanoparticles’ surface ultimately depends on the intermolecular forces, which dictate the mutual interaction transforming their physical, chemical, and biological characteristics. Therefore, a better understanding of the adsorption of serum proteins and their impact on nanoparticle physicochemical properties is of utmost importance for developing nanoparticle-based therapies. We investigated the interactions between potentially therapeutic proteins, neurotrophin 3 (NT3), brain-derived neurotrophic factor (BDNF), and polyethylene glycol (PEG), in a cell-free system and a retinal pigmented epithelium cell line (ARPE-19). The variance in the physicochemical properties of PEGylated NT3–BDNF nanoparticles (NPs) in serum-abundant and serum-free systems was studied using transmission electron microscopy, atomic force microscopy, multi-angle dynamic, and electrophoretic light scattering. Next, we compared the cellular response of ARPE-19 cells after exposure to PEGylated NT3–BDNF NPs in either a serum-free or complex serum environment by investigating protein release and cell cytotoxicity using ultracentrifuge, fluorescence spectroscopy, and confocal microscopy. After serum exposure, the decrease in the aggregation of PEGylated NT3–BDNF NPs was accompanied by increased cell viability and BDNF/NT3 in vitro release. In contrast, in a serum-free environment, the appearance of positively charged NPs with hydrodynamic diameters up to 900 nm correlated with higher cytotoxicity and limited BDNF/NT3 release into the cell culture media. This work provides new insights into the role of protein corona when considering the PEGylated nano–bio interface with implications for cytotoxicity, NPs’ distribution, and BDNF and NT3 release profiles in the in vitro setting.

## 1. Introduction

Neurotrophic factors have been identified as potentially therapeutic agents owing to their protective properties for photoreceptors in models of retinal and neuronal degeneration [1,2,3]. Neurotrophins (NTs), including brain-derived neurotrophic factor (BDNF) and neurotrophin-3 (NT3), bind to their tropomyosin-related receptor kinase (TrkB and TrkC) and promote neuronal differentiation, survival, and plasticity [4,5]. Although NTs are promising neuroprotective agents, their short half-life impairs the efficacy and mode of action at the target site [6,7]. Advancements in nanoparticle delivery and protein encapsulation within biodegradable materials could significantly improve the intraocular biocompatibility and retention time of NTs [8]. However, an in vivo environment limits nanoparticles’ (NPs) bioavailability. One of the main concerns is the complexity of the serum components (such as albumin, apolipoproteins, complement system, and immunoglobulins) and their effect on NP–target cell interactions [9]. The adsorption phenomena of biological layers on NPs’ surface, referred to as protein corona, alter the kinetics and physicochemical properties of NPs at their target destination [10]. Understanding this process is crucial for ocular drug delivery, where the electrical potential of the vitreous body strongly affects the fate of NPs in the eye [11]. In such a complex protein environment, ligands with lower binding affinity could dominate the NP surface, reducing successful diffusion, cellular internalization, and distribution. Hence, the functionality and targeting of NPs is dictated by their design, formulation, and surface charge.

Polyethylene glycol (PEG) is extensively used as a “stealth” polymer to formulate nanoparticles owing to its safety in humans and the fact that it is FDA-approved [12,13]. Some researchers observed that biocompatibility can be significantly improved after PEG adsorption at nanoparticles’ surface by the PEG layers’ self-assembled arrangement, depending on PEG chain lengths and densities. The unsaturated PEG layer on the NP surface, controlled by the size and steric interactions of surface-bound PEG, would lead to 2D molecular sieving of proteins with a distinct size selectivity when the dimensions of the proteins and PEG are similar [14]. Furthermore, PEG properties of high hydrophilicity and low in vivo toxicity prolong NPs’ half-life in circulation by limiting electrostatic and hydrophobic interactions between NPs and macrophages, resulting in lower uptake in the reticuloendothelial system and clearance without reaching their target cells [4,5,6]. It has also been reported that PEGylated NPs are less prone to the formation of protein corona after administration, which is crucial for successful diffusion throughout the vitreous cavity, penetration into the retinal layers, and interaction with the constituent layer, including endothelial and neuronal cells [2]. In addition, nanoparticles encounter a complex interplay with the innate immune system upon entering the eye. Opsonization, a pivotal component of this interaction, involves the binding of opsonins (e.g., antibodies, complement proteins, C-reactive protein, and mannose-binding lectin) to nanoparticle surfaces, thus marking them for rapid phagocytosis by immune cells. The synthesis of nanoparticles employing PEG enables the prolonged delivery of therapeutic drugs by avoiding the opsonization process and uptake by the mononuclear phagocytic system (MPS). Consequently, the reduction in inflammatory effects improves the payload delivery and the drug’s therapeutic efficacy [7,8]. Moreover, the interaction of NPs with biological fluids immediately alters nanoparticles’ surface properties, e.g., the surface charge distribution, the zeta potential, and the accessibility of several chemical functional groups, consequently defining the stability of NPs in suspension, as well as their aggregation and biological response.

In relation to ocular diseases such as age-related macular degeneration (AMD), there is still room for improvement to overcome the difficulties of successful delivery of therapeutics to the retinal cells through viscous vitreous fluid [9,10]. Previously, human retinal pigmented epithelial cells (ARPE-19) were used as a model cell line to study the effect of NPs [12,13]. However, those studies investigated mostly NPs with encapsulated, already-in-use drugs and induced high cytotoxicity owing to their physicochemical properties. Therefore, we used ARPE-19 cells in the current study to understand how novel, biodegradable NPs with two adsorbed neurotrophins in one nanocarrier affect cell viability and morphology.

Our study comprehensively characterized PEGylated NT3–BDNF nanoparticles in a serum-free and complex serum environment. Fetal bovine serum (FBS) was used to simulate the wide array of proteins in vivo and the protein corona formation. First, we focused on determining PEGylated NT3–BDNF NPs’ stability in cell-free systems and then in retinal (ARPE-19) cells.

The influence of serum exposure on the spontaneous self-assembly of PEGylated NT3–BDNF complexes with different NT3/BDNF concentrations, hereinafter referred to as “PEGylated NT3–BDNF nanoparticles”, was studied in relation to physicochemical properties, using atomic force microscopy (AFM), transmission electron microscopy (TEM), and multi-angle dynamic and electrophoretic light scattering (MADLS/ELS). Next, we determined if the serum exposure modified BDNF/NT3 release in a cell-free system in regard to variations in the functionality of NPs. Finally, we explored the in vitro interactions between NPs and ARPE-19 cells, studying retinal cell morphology, cytotoxicity, and the ability for cellular internalization after 72 h.

To the best of our knowledge, this is the first time we have reported data regarding the mechanism of self-assembly of two neurotrophins and PEG chains that appeared to be clearly organized nanocarriers. Moreover, we thoroughly described the influence of protein corona adsorption on biocompatible PEGylated NT3–BDNF nanoparticles in in vitro settings, consequently delivering a well-characterized nanocarrier system for potential nanomedical implications.

## 2. Materials and Methods

### 2.1. Preparation of PEGylated NT3–BDNF Nanoparticles

#### 2.1.1. BDNF and NT-3

In our studies, unfiltered stock solutions of carrier-free recombinant human BDNF (rhBDNF) (248-BDB-250/CF, R&D Systems, Minneapolis, MN, USA) as well as carrier-free recombinant human NT3 (rhNT3) (248-N4-250/CF, R&D Systems), of known concentrations (typically 250 mg L^−1^), were prepared in phosphate buffered saline (PBS) pH 7.4 ± 0.2, 0.15 M (Biomed, Lublin, Poland) and stored for no longer than 2 months at a temperature of −20 °C. rhBDNF and rhNT3 are hereinafter collectively referred to as neurotrophins (NTs). Before each measurement, the stock solution was diluted to the desired bulk concentration, typically 25 mg L^−1^. The exact concentration of these solutions was determined by the commercially available enzyme-linked immunosorbent assay (ELISA) assay (DY992, DY990, DY994, DY999, DY995, WA126, DY006, DY268, R&D Systems). The temperature of the experiments was kept at a constant value equal to 298 ± 0.1 K.

#### 2.1.2. PEG

PEG (poly (ethylene glycol)) of high analytical grade was used without further purification. PEG with a molecular weight of 4 kDa (1546569, Sigma Aldrich, St. Louis, MO, USA) was used for simultaneous encapsulation of rhBDNF and rhNT3. PEG working solution was prepared by weighing out 1 g of PEG, dissolving it in 10 mL of PBS solution, and gently rotating the tube up and down for 15 min at room temperature. After dissolution, the solution was filtered through a 0.22 µM filter.

#### 2.1.3. PEGylated NT3–BDNF Nanoparticles

PEGylated NT3–BDNF nanoparticles containing rhBDNF and rhNT3 proteins were prepared in three different concentrations of 0.1, 1, and 5 mg L^−1^. The process was carried out at pH 7.4, an ionic strength of 0.15 M, and at room temperature. The adsorption of neurotrophins was successfully conducted in a PEG solution at a concentration of 10,000 mg L^−1^. The PEGylation process of NTs was carried out in the following manner: firstly, the electrophoretic mobility of PEG suspension at a concentration of 10,000 mg L^−1^ was determined. Secondly, rhNT3 was PEGylated at a specific concentration by mixing rhNT3 stock solutions with PEG suspension to create three different types of PEGylated NT3 nanoparticles formulated with 0.1, 1, or 5 mg L^−1^ of rhNT3. The solutions were mixed at room temperature for 15 min. Then, the electrophoretic mobility of all three PEGylated NT3 conjugates was measured. Next, PEGylated NT3–BDNF conjugates were made by mixing PEGylated NT3 suspension with rhBDNF solution to obtain PEGylated NT3–BDNF conjugates with 0.1, 1, or 5 mg L^−1^ concentrations of both neurotrophins. Finally, the electrophoretic mobility of the PEGylated NT3–BDNF conjugates at 0.1, 1, and 5 mg L^−1^ BDNF and NT3 was measured. The adsorption of proteins at the polyelectrolyte chains led to the formation of PEGylated NT3–BDNF nanoparticles through spontaneous self-assembly. The obtained highly efficient complexes are referred to as “PEGylated NT3–BDNF nanoparticles”.

#### 2.1.4. FITC-PEGylated Neurotrophin Nanoparticles

To detect PEGylated NT3–BDNF nanoparticles in ARPE-19 cells by confocal microscopy, we used FITC-PEG-NHS (Biopharma PEG Scientific Inc., Water-town, MA, USA). Fluorescein-labeled PEGylated NT3–BDNF nanoparticles containing rhBDNF and rhNT3 in three concentrations of 0.1, 1, and 5 mg L^−1^ were prepared as described in Section 2.1.3. FITC-PEG-NHS esters were used instead of PEG as a fluorescent label for labeling NT3–BDNF complexes. After transferring the appropriate volume of protein solution related to 0.1, 1, and 5 mg L^−1^ to the vial containing the NHS-PEG-FITC, the suspensions were mixed well and incubated in a thermoshaker at room temperature for fifteen minutes to allow the reaction between succinimidyl esters and primary amines (RNH2). The 24 h dialysis procedure was performed, which involved the use of Pur-A-Lyzer 6000 tubes to eliminate unconjugated dye and polyelectrolyte and purify fluorescently labeled FITC-PEGylated NT3–BDNF NPs. The process continued until the absorbance of the filtrate at 495 nm reached the background levels. After measuring the UV/VIS spectrum of conjugates, the degree of labeling (DOL) of proteins was determined by applying the following equation:(1)$$\mathrm{DOL}=\frac{A_{max}\times \varepsilon_{280}}{\left(A_{280}\times A_{max}\times CF\right)\times \varepsilon_{max}}$$
where *A_max_* and *A*_280_ are absorbances of the conjugate solution measured at 495 nm (λ_max_ of the dye) and 280 nm, respectively. The absorbance of FITC indicated the amount of dye present in the conjugates, while the absorbance of the proteins at 280 nm provided the total protein concentration. *ε*_280_ is the extinction coefficient of protein (for rhBDNF and rhNT3, 22.648 M^−1^ cm^−1^ and 29.638 M^−1^ cm^−1^, respectively, were used). *ε_max_* is the extinction coefficient of the dye at the absorption maximum equal to 71.000 M^−1^ cm^−1^. *CF* (0.11) is the correction factor, which is required to eliminate the contribution of the dye at 280 nm.

### 2.2. Physicochemical Characterization of PEGylated NTs’ Nanoparticles

#### 2.2.1. Transmission Electron Microscopy (TEM)

A JEOL JSM-7500F electron microscope working in the transmission mode (TEM) was used to evaluate the morphology and size distribution of PEGylated NT3–BDNF nanoparticles’ suspension (in 0.15 M PBS). The micrographs were analyzed using MultiScan 6.08 software (A computer scanning system). All images represent detection directly from the sample surfaces, with no coating or contrasting applied. The size of PEGylated NT3–BDNF nanoparticles with 0.1, 1, and 5 mg L^−1^ adsorbed rhBDNF and rhNT3 proteins was determined using ImageJ software (https://imagej.net/ij/index.html (accessed on 15 August 2023)) by gathering the number and coordinates of single nanoparticle molecules. Manual counting of PEGylated NT3–BDNF nanoparticles was based on comparing the original image and the same picture altered by digital image filters by cutting off the picture background. The histograms of PEGylated NT3–BDNF nanoparticles were generated from the analysis of a minimum of 300 NPs.

#### 2.2.2. Atomic Force Microscopy (AFM)

Samples for AFM were prepared from PEGylated NT3–BDNF nanoparticles with 0.1, 1, and 5 mg L^−1^ adsorbed BDNF and NT3 proteins’ suspensions (in 0.15 M PBS) by immersing a mica sheet in 2000 μL of the suspension without the surface of mica modification. The solid pieces of mica were freshly cleaved to thin sheets before every experiment. After 15 min of deposition, mica was carefully rinsed with distilled water to remove any trace of solute that could crystallize at the support surface. The samples were left for air-drying until the next day. Next, a prepared sample was positioned on the holder for AFM scanning. Sizes and morphologies of phage particles were investigated by AFM imaging in air using the NT-MDT device. All measurements were performed in the semi-contact mode using high-resolution silicon probes (NSC35/AlBS, MicroMasch, Sofia, Bulgaria), and the cone angle of the tip was less than 20°. The images were recorded at a scan rate of 1 Hz for the randomly chosen places. The images were flattened using an algorithm provided with the instrument.

#### 2.2.3. Size Measurement of PEGylated NTs with Multi-Angle Dynamic Light Scattering (MADLS)

To determine the total concentration of PEGylated NT3–BDNF nanoparticles, we measured the particle size distribution, time-averaged intensity scattered by a molecular scatterer, and the sample using multi-angle dynamic light scattering. MADLS is a well-established technique for determining the hydrodynamic size distribution of molecules or NPs dispersed in solutions. The multi-angle detection provides reliable data for relatively polydisperse samples, which improves particle sizing resolution, sensitivity, and accuracy compared with traditional single-angle dynamic light scattering. Particle concentration measurements were carried out for the bulk of the size and the concentration of the PEGylated NT3–BDNF nanoparticle. Afterward, we used the algorithm derived from scattering intensity information to calculate the particle concentration. Suspensions of nanoparticles, PEG, NT3, and BDNF, diluted with PBS buffer to a suitable concentration were measured using DLS and MADLS, as previously described [15,16,17]. Briefly, the sizes of molecules were measured in the Zetasizer Ultra apparatus (Malvern Instruments, Malvern, UK) equipped with a laser of 633 nm wavelengths and an He-Ne laser at a maximum power of 10 mW. Size distribution was obtained from diffusion coefficients recalculated to the size by assuming the spherical shape of particles [11]. The obtained values represent the diameter of spherical particles, which move in viscous media with the same velocity as protein, PEG polyelectrolyte, or PEGylated NT3–BDNF nanoparticles. All experiments were performed at 25 °C using a sample volume of 1 mL and disposable cuvettes (DTS0012, Malvern Panalytical Ltd., Malvern, UK). The instrument settings were optimized automatically using the ZS XPLORER 3.2.0 software (Malvern Panalytical Ltd., UK). The precision of the MADLS concentration measurements was evaluated from 10 repeated measurements with an acquisition time of approximately 200 s each. Each size measurement was presented as the average value of 20 runs, with triplicate measurements within each run. Samples were measured directly after preparation; left at room temperature; and measured again after 30 min, 2 h, and 72 h.

#### 2.2.4. Nanoparticle Zeta (ζ) Potential Determination

A Zetasizer Ultra instrument was also used to assess the stability of NPs in the suspension under various NTs’ concentrations and conditions with or without FBS at the temperature of 37 °C through diffusion coefficients (D) and electrophoretic mobility (μe) measurements. The electrophoretic mobility of BDNF, NT3, PEG molecules, and PEGylated NT3–BDNF nanoparticles was measured at PBS, PBS supplemented with 1% FBS, and 10% FBS with laser Doppler velocimetry (LDV) technique with the aid of a Malvern device. The LDV method was introduced by Adamczyk et al. and is based on the measurement of ζ-potential/micro electrophoretic mobility changes during the adsorption of the tested protein on a model colloid particle [16]. Electrophoretic mobility was recalculated to ζ-potential using the Henry equation, which is valid for higher ionic strength, where the polarization of the electric double layer is relevant (the double-layer thickness becomes smaller than the protein dimension).

### 2.3. Cell Culture

The human retinal pigment epithelial (ARPE-19) cells were purchased from American Type Culture Collection (ATCC, Rockville, MD, USA). The cells were cultured in a t-75 NUNC cell culture flask with Dulbecco’s Modified Eagle Medium/Nutrient Mixture F-12 (1:1) growth medium (DMEM/F-12) received with L-glutamine, 4.5 g L^−1^ D-glucose, and pyruvate (Life Technologies, Paisley, UK) containing 10% inactivated fetal bovine serum (FBS; Life Technologies) penicillin–streptomycin (100 U/mL–100 µg/mL; Life Technologies) at 37 °C in a humidified environment of 5% CO_2_ and were allowed to grow for 7 days. Cultures were maintained with weekly subculture using Trypsin-EDTA 0.25% (Life Technologies) and fed every 2 to 3 days. Cells from passages 18 to 20 were used in the cytotoxicity, rhBDNF/rhNT3 release, and nanoparticle internalization studies. Cells were used after they reached 90% confluence.

### 2.4. Cytotoxicity Assay

We assessed the effect of nanoparticles on cell viability with a colorimetric assay, alamarBlue^®^ (Life Technologies). The alamarBlue assay measures the fluorescence value at the emission peak of resorufin. ARPE-19 cells were harvested using Trypsin-EDTA 0.25% (Life Technologies) and seeded at a density of 10 k cell/well in 96-well plates in medium without phenol red and various serum conditions; DMEM/F-12 with 10% FBS (referred to as complex media) and DMEM/F-12 with 1% FBS, both supplemented with 1% penicillin–streptomycin (100 U/mL–100 μg/mL). The cells were incubated at 37 °C in a humidified environment of 5% CO_2_ for 24 h. After 24 h in serum-supplemented media, the culture medium was removed and cells were re-incubated with DMEM/F-12 with either (1) 10% FBS, (2) 1% FBS, or (3) serum-free, all supplemented with 1% penicillin–streptomycin (100 U/mL–100 μg/mL).

Moreover, cells were exposed for 30 min and 72 h to PEGylated NT3–BDNF nanoparticles with 0.1, 1, and 5 mg L^−1^ adsorbed BDNF and NT3. The ratio of administered PEGylated NT3–BDNF NPs and medium was one to four, respectively. Untreated cells (in cell culture medium with various serum concentrations—serum-free, 1% and 10% FBS) were used as a negative control. After each time point, the medium was aspirated and 100 μL of 10% alamarBlue prepared with 1% FBS DMEM/F-12 was added to cells. The 96-well culture plate was then incubated at 37 °C and 5% CO_2_ for 5–6 h. The absorbance was read at 560/590 nm using a Varioskan LUX Multimode Microplate Reader (Thermo Fisher, Waltham, MA, USA). Twelve replicates were used for each treatment. The cell viability was calculated as a percentage of non-treated cell viability.

### 2.5. In Vitro BDNF and NT3 Release Profile Quantification in Serum-Free and Serum-Rich Environments

#### 2.5.1. Release Profile Quantification and Nanoparticle Size Distribution in a Cell-Free System

All of the release studies in a cell-free environment were conducted under sink conditions as a no sink condition could substantially underestimate the drug release from nanoparticles [18]. The determination of encapsulated neurotrophins’ release over time was performed by preparing PEGylated NT3–BDNF nanoparticles as in Section 2.1. The release of rhBDNF and rhNT3 was studied in PBS and 1% and 10% FBS. Prepared nanoparticle solutions were continuously mixed at 200 RPM at 37 °C (Thermo-Shaker, Model MSC-100, ABChem, Dorval, QC, Canada). Samples were then taken at time 0 (15 min after preparation of nanoparticles), after 30 min, and after 72 h, and subsequently replaced with either PBS or 1% or 10% FBS. The samples were immediately ultracentrifuged at 60,000 RPM at 4 °C for 40 min. Supernatants were harvested and frozen at −20 °C. The released rhBDNF/rhNT3 concentration levels were determined using enzyme-linked immunosorbent assay (ELISA). ELISA kits were purchased from R&D Systems (Minneapolis, MN, USA) as the Duo Sets containing capture and secondary antibody, protein standard, and Streptavidin-HRP. The concentration was calculated using UV/VIS. The relative percent of released rhBDNF/rhNT3 was determined in relation to the adsorbed NTs onto PEGylated NT3–BDNF nanoparticles. Thus, the detected rhBDNF/rhNT3 concentrations at time 0 were considered as not adsorbed to the PEG surface. Hence, this concentration was subtracted from the original PEGylated NT3–BDNF concentration (either 0.1, 1, or 5 mg L^−1^). In parallel, the presence of nanoparticles in the supernatant was checked using the MADLS method to confirm the reliability of this method.

#### 2.5.2. In Vitro Release Profile in Cell Culture

The release of rhBDNF/rhNT3 was determined using ELISA. The concentration of released NTs was determined by exposing ARPE-19 cells to various serum and PEGylated NT3–BDNF nanoparticle concentrations, as described in Section 2.4. After each time point (30 min and 72 h), the cell supernatant was collected to quantify rhBDNF/rhNT3 concentration using UV/VIS spectroscopy calculated according to the ELISA standard curve and manufacturer’s protocol. The relative concentration of released NTs was determined in relation to the adsorbed rhBDNF/rhNT3 onto PEGylated NT3–BDNF nanoparticles.

### 2.6. Confocal Microscopy

To visualize morphological changes and nanoparticle internalization, ARPE-19 cells were seeded in a Nuncä Lab-Tekä 4-well Chamber Slide System (Thermo Fisher) at a density of 50,000 cells/well and incubated for 24 h at 37 °C and 5% CO_2_ in cell culture medium. Cells were exposed to various serum and PEGylated nanoparticle concentrations as described in Section 2.4. Cells were fixed with 4% formaldehyde for 15 min and washed with PBS. For cell membrane staining, the slides were incubated with 1:50 Wheat Germ Agglutinin-Tetramethyl rhodamine, which binds to N-acetyl-glucosamine and N-acetylneuraminic acid (sialic acid) residues on the surface of cell membranes (Thermo Fisher) in HBSS buffer for 20 min. After washing, the slides were counterstained with 4′,6-diamidino-2-phenylindole (DAPI), followed by PBS rinse, and mounted with VECTASHIELD^®^ mounting medium for fluorescence (Vector Laboratories, Inc., Burlingame, CA, USA). Slides were stored at 4 °C until the examination. Visualization was performed with fluorescent microscope (Zeiss LSM-700, Oberkochen, Germany) using a 20× objective. All presented images were presented with 3× magnification inserts.

### 2.7. Statistical Analysis

Data from each experiment are presented as the mean ± standard error of the mean (SEM). Treated groups were compared to untreated control. Statistical tests were performed using GraphPad Prism 9 (GraphPad Software, Inc., CA, USA). Two-way ANOVA with post-hoc Tukey’s test was performed to compare diverse groups of various treatments. In all analyses, a *p*-value ≤ 0.05 was considered significant.

## 3. Results

### 3.1. Physicochemical Characterization of PEGylated NT3–BDNF Nanoparticles with Various Neurotrophins’ Concentrations in PBS

We obtained topographical scans of immobilized nanoparticles using atomic force microscopy and transmission electron microscopy (TEM). These methods allow complete characterization at nanometer resolution and allow a reliable determination of the nanoobject’s size by studying dispersion/aggregation processes for the PEGylated NT3–BDNF complexes [8,19]. The imaging demonstrated that the size distributions of PEGylated NT3–BDNF nanoparticles were dependent on the rhBDNF and rhNT3 concentrations. By increasing the neurotrophin concentration in the PEG suspension, increased clustering between PEG and NTs was observed (Figure 1 and Figure 2). The diameter of PEGylated NT3–BDNF nanoparticles with different concentrations of both neurotrophins (0.1, 1, and 5 mg L^−1^) was measured under higher magnification using both TEM (Figure 1A–C, Appendix A) and AFM (Figure 2 and Appendix A). At 5 mg L^−1^, the size distribution of PEGylated NT3–BDNF NPs shifted to an average diameter of 50 nm, indicating the process of NPs ungrouping.

Using AFM, we could visualize an average nanoparticle diameter adsorbed under the diffusion-controlled transport condition at the mica surface. The results obtained in this work correlated with our previous studies that revealed the adsorption of neurotrophin aggregates under the diffusion-controlled transport condition on the mica surfaces [20]. Here, the PEGylated NT3–BDNF nanoparticles were also adsorbed onto the mica surface as separate objects with an average diameter of up to 10 nm for 0.1 and 1 mg L^−1^ neurotrophin concentrations. The distribution of NPs changed to non-uniform aggregates on the mica surface and showed highly heterogeneous forms, varying in shapes and sizes regarding neurotrophin concentration.

Subsequently, we determined the size, polydispersity index (PDI), concentrations, and zeta potential of PEGylated NT3–BDNF nanoparticles with different amounts of adsorbed recombinant neurotrophins—based on the MADLS and electrophoretic light scattering (ELS) studies.

The average neurotrophins’ sizes determined using MADLS, when particles existed in a native hydration state, corresponded to values from the TEM study (Figure 3C,D). Neurotrophins’ aggregation in solution with low ionic strength, such as PBS, appeared to be a highly dynamic process, causing significant changes in the distribution of individual biomacromolecules. The typical size of the rhBDNF molecule was 301.7 nm (with PDI equal to 1.2) and that of the rhNT3 molecule was 3.03 nm (with PDI equal to 0.8).

After synthesis, the average size of PEGylated NT3–BDNF NPs was 3.71 nm at 10^17^ particles/mL for all used concentrations of rhNT3 and rhBDNF (Appendix A). The polydispersity index of NPs decreased unequivocally in relation to rhBDNF or rhNT3, reaching 0.26, 0.4, and 0.6 for NPs’ neurotrophin concentrations of 0.1, 1, and 5 mg L^−1^, respectively. In addition, we observed a peak for PEG, with an intensity of ~20%, which decreased after neurotrophins’ adsorption to ~10%. This may indicate that the adsorption of PEG chains on the NTs’ surface could be caused by weak electrostatic interactions.

The stability of PEGylated NT3–BDNF complexes was measured by determining the hydrodynamic size as a function of time and rhBDNF/rhNT3 concentration. MADLS analysis indicated that spontaneous self-assembly of PEGylated NT3–BDNF complexes with similar morphology and hydrodynamic diameter, with an average size of 10 nm (10^17^ particles/mL) by the number, was the most common (Figure 3B). Kinetics studies indicated that suspensions of PEGylated NT3–BDNF NPs had four groups of particles with hydrodynamic diameters between 3.5 and 930 nm after 72 h. In contrast, after 2 h, NPs had three or two groups of particles with different hydrodynamic diameters, between 3.5 nm and 430.6 nm, with a polydispersity index lower than 0.4.

Afterwards, the PEGylated NT3–BDNF nanoparticles were physiochemically characterized in regard to electrophoretic mobility/zeta potential. The PEG adsorption on positively charged rhNT3 (5 ± 2.5 mV) and slightly negatively charged rhBDNF (−2 ± 2.5 mV) continued according to the bulk diffusion transport. Irrespectively of the initial PEG chain charge (−4 ± 2 mV), PEGylation of rhNT3 (5 mg L^−1^) significantly increased the zeta potential of the entire system. The PEGylated NT3 complexes, comprising 0.1 mg L^−1^ rhNT3, were positively charged (3 ± 1.2 mV). Whereas, after rhBDNF adsorption, the net electrokinetic charge decreased to −10.8 ± 4.2 mV. Further, the experimental data indicated that the electrokinetic charge of PEGylated NT3–BDNF nanoparticles became more positive over time for all used protein concentrations after 72 h, eventually reaching 1.9 mV (for BDNF/NT3 at 5 mg L^−1^) (Table 1, Figure 4).

### 3.2. Physicochemical Characterization of PEGylated NT3–BDNF Nanoparticles with Various Neurotrophin Concentrations in FBS

We used PBS supplemented with different FBS concentrations to assess the aggregation/dispersion process of the PEGylated NT3–BDNF NPs. Those conditions mimicked the formation of the protein corona in vitro. Here, the kinetics of PEGylated NT3–BDNF nanoparticles was determined by characterizing the hydrodynamic size, PDI, NP concentrations, and zeta potential as a function of time.

MADLS analysis of positively and negatively charged PEGylated NT3–BDNF nanoparticles revealed that the number of NPs with diameters larger than 100 nm decreased with FBS supplementations and attained a low PDI of less than 0.45 (Appendix A). As shown in Figure 5, the number of NPs (0.1 mg L^−1^ NTs) with a hydrodynamic diameter of 431 nm decreased from 12% in PBS to 0% in PBS supplemented with 10% FBS, after 30 min. The number of NPs with diameters greater than 100 nm also decreased with increased FBS concentration for 1 mg L^−1^ and 5 mg L^−1^. Furthermore, the percent intensity of nanoparticles with diameters smaller than 100 nm increased over time for any given solution (e.g., PBS, 1% FBS, and 10% FBS). This relationship applies to all NPs, regardless of BDNF/NT3 concentration.

Additionally, microelectrophoresis measurements revealed that PEGylated NT3–BDNF nanoparticles suspended in PBS gained a negative zeta potential (Table 1), increasing with time and NT3/BDNF concentrations. In contrast, the electrokinetic charge of PEGylated NT3–BDNF nanoparticles reached an average value of approximately −10 ± 2 mV after incubation in either PBS with 1% FBS or 10% FBS, regardless of the neurotrophin concentrations and exposure time used. Incubation of NPs with serum resulted in a “normalization” of the zeta potential due to the adsorption process. The zeta potential of PBS supplemented with 1% FBS or 10% FBS solutions without nanoparticles was −11 mV or −13 mV, respectively. After 72 h of incubation, as expected, FBS significantly decreased the negative zeta potential of PEGylated NT3–BDNF nanoparticles compared with NPs dispersed in PBS.

### 3.3. Cell Viability Study in Different Media Conditions

Various serum and PEGylated NT3–BDNF NPs’ concentrations can have a different effect on cell viability. Thus, ARPE-19 cells were simultaneously exposed to (1) rhBDNF and rhNT3 released from PEGylated nanoparticles and (2) DMEM/F-12 supplemented with 10% FBS (referred as complex media), 1% FBS, or serum-free, for up to 72 h. NPs in concentrations ranging from 0.1 mg L^−1^, 1 mg L^−1^, and 5 mg L^−1^ did not show a concentration-dependent effect on ARPE-19 viability after 30 min of incubation, as no significant cytotoxicity effect was noticed. However, a time- and serum-dependent cytotoxic effect was observed for all concentrations of NPs after 72 h of incubation. Figure 6 and Appendix A show a significantly lower ARPE-19 viability in serum-free media, yielding ~70%, ~65%, and 60% for 0.1 mg L^−1^, 1 mg L^−1^, and 5 mg L^−1^, respectively. Overall, in complex media, we did not see any significant decrease in cell viability at any given NP concentration and incubation time.

### 3.4. rhBDNF and rhNT3 Release Study under Sink Conditions in a Cell-Free System

PEGylated NT3–BDNF NPs released both rhNT3 and rhBDNF in a time-dependent manner. Figure 7 represents the release profiles in PBS and PBS supplemented with 1% and 10% FBS with the cumulative release percentage. In the serum-free buffer, the initial release in the first 30 min reached ~100% of the adsorbed neurotrophins (Figure 7A). The amount of released rhNT3 in PBS was greater by ~50% than in PBS supplemented with 10% FBS. At 0.1 mg L^−1^, the release kinetics of rhNT3 plateaued after 30 min. Meanwhile, the release kinetics of the nanoparticles in PBS supplemented with 1% FBS decreased from ~75% to ~10% from 30 min to 72 h, respectively. At 5 mg L^−1^, a similar release rate was observed. Interestingly, rhBDNF release remained at 70% for any used neurotrophin concentrations and buffers after 30 min and 72 h (Figure 7B).

### 3.5. In Vitro NT3/BDNF Release Profile from PEGylated Nanoparticles in Different Media Conditions

PEGylated NT3–BDNF NP release kinetics was further investigated using the ARPE-19 cell line (Figure 8). In the biological milieu, the highest rhNT3 and rhBDNF release was in complex medium. PEGylated NT3–BDNF nanoparticles at 0.1 mg L^−1^ did not show a significant release concentration of rhNT3. Although the rhNT3 concentration peaked after 30 min of incubation at ~50 pgmL^−1^ (in a complex medium), it decreased after 72 h to ~3 pgmL^−1^. At 5 mg L^−1^, we observed a significant, time-dependent increase in rhNT3 concentration over 72 h. Complex media caused a significant increase in rhNT3 release, reaching ~5-fold serum-free levels. At 1% FBS conditioned medium, rhNT3 concentration was ~3-fold greater than in serum-free medium after 30 min. The release rate of rhBDNF from PEGylated NT3–BDNF NPs (0.1 mg L^−1^) was significantly decreased after 72 h. The concentration of rhBDNF decreased by ~3-fold and ~5-fold when incubated with 1% and 10% FBS supplemented medium, respectively. No significant release was noticed in serum-free media (Figure 8B). At 5 mg L^−1^ BDNF/NT3, we did not notice any significant changes in BDNF release profile in serum-free and complex media after 72 h, similarly to 1 mg L^−1^ (Appendix A and Figure 8B). In comparison, in 1% FBS, the level of rhBDNF decreased significantly by ~2-fold from 30 min to 72 h (Figure 8B).

### 3.6. Confocal Microscopy

We used confocal microscopy to elucidate further how different nanoparticle concentrations and serum abundance affect ARPE-19 cytotoxicity. The increase in serum concentration in the cell media did not cause significant changes in the ARPE-19 morphology and cytotoxicity after 30 min of incubation (Appendix A).

When cells were grown in complex media, cell stress was not observed. The cytotoxic effect was significant in serum-free media after 72 h, regardless of PEGylated NT3–BDNF concentration. Incubation for 72 h showed that cells in serum-free conditions and in medium supplemented with 1% FBS presented with a decreased cell number for all tested concentrations of neurotrophins in comparison with the control (Figure 9).

The cellular internalization of PEGylated NT3–BDNF NPs was noted after 72 h when compared with 30 min (Figure 9 and Appendix A). After 72 h, the fluorescence was mostly localized to the cytosolic space rather than the plasma membrane.

## 4. Discussion

Despite continuous efforts to study therapeutic protein adsorption on a wide array of biodegradable nanomaterials, this phenomenon is not fully understood, thus posing limitations when it comes to the design and formulation of nanomaterials for biological and medical purposes. Although several studies focused on assessing protein corona formation on colloidal carriers after incubation in complex biological fluids such as serum [19,21], human plasma [20,22], or cell culture media [23,24], there are only a few that describe the interactions with biodegradable NPs. Additionally, research on NPs with encapsulated therapeutic proteins is limited, and a thorough understanding of the physicochemical characteristics dependent on protein corona is scarce [25,26,27,28]. For protein-based nanoparticles, it is crucial to determine the interactions (noncovalent hydrogen bonds, electrostatic and hydrophobic effects) between nanocarriers and proteins to improve strategies for limited protein binding and release [6,29,30,31].

We formulated PEGylated neurotrophin-based nanoparticles and evaluated their interplay both in cell-free systems and with the ARPE-19 cell line. The main research hypothesis assumes that the range of interactions within nanocarriers is modulated not only by the physicochemical properties, but also by the composition of the biological environment.

PEG is a hydrophilic ligand that tends to be colloidally stable in a wide range of solvents with high ionic strength, including biological fluids [32,33]. The influence of PEG on the adsorption of proteins can vary as a function of the molecular weight [14,34] and polymer chain architecture, where the surface density of PEG is typically the most critical parameter [35]. In the present study, firstly, we focused on the electrokinetic charge of NTs in constructing PEGylated NT3–BDNF nanoparticles using an approach where adsorption between NTs and PEG molecules is mediated through electrostatic coupling and transient hydrogen bonding. A similar approach was presented for another nanoscale complex of protein polymers; e.g., PEG-PLE with BDNF protein, PAMAM dendrimers with NT-4, and PEGylated neurotrophin-based PAMAM nanoparticles, where binding between NTs and polymers were mediated in electrostatic ways [14,16,34,36,37,38,39,40]. We carried out BDNF/NT3 adsorption on PEG (having a low molecular weight) in a high ionic strength solution (0.15 M), reducing the repulsive interactions between the protein and polymers that have been previously reported [37,38,39,40,41]. Our results are in good agreement with profiles of protein adsorption where smaller proteins possess larger curvature that reduces the lateral interaction at PEG with low grafting chain densities, thus favoring the ”mushroom” conformation [42,43]. In the present study, experimental observation indicates that PEG-NT3/BDNF interactions are preferred (Figure 1, Figure 2, Figure 3 and Figure 4); the zeta potential of the PEGylated NT nanoparticles was much lower than the zeta potential of bulk protein, which could correspond to the formation of an unsaturated BDNF/NT3 layer at PEG chains causing structural alterations/conformation change to NTs that can be induced upon binding.

The PEGylated NT3–BDNF NPs were exposed to varied concentrations of serum in PBS to investigate the effect of protein corona on their size, zeta potential, aggregation state, and exposure time. We found that, after serum incubation, the average size of nanoparticles was significantly decreased to ca. 100 nm. This reflects the dispersion (regrouping) process for all three studied NT3/BDNF concentrations under serum conditions. However, we noticed a higher aggregation (grouping) rate and greater NP size in the serum-free conditions. This strongly indicates the reversibility of the process and its dependence on serum supplementation [19,20,22,28,44,45,46]. Further, the kinetics of the serum protein’s adsorption at NPs is closely related to the specific type of nanoparticle, surface properties, size, composition of the serum proteins, and the local environment. These factors play a significant role in determining the kinetics of the nanoparticle–serum protein complex, e.g., the size of NPs depends on the exposure time to the serum protein’s environment [11,18,21]. Here, we observed that significant variations in size occurred not only within 30 min, but also after 72 h of incubation with serum proteins for all examined nanoparticles, which was in good accordance with other studies that showed a time-dependent effect on both NP size and protein corona composition [25,29,30,31].

Prior to the formulation of PEGylated NT3–BDNF nanoparticles, we focused on the determination of the electrokinetic charge of the rhNT3, rhBDNF, and PEG chain to further study the adsorption phenomena and NPs’ functionality. It has been reported elsewhere that differently charged NPs, irrespective of the initial surface functionalization, display an overall negative charge after protein corona exposure [47,48]. Contrary to our study, Palchetti et al. pointed out that incubation with complex serum resulted in a “normalization” of NPs’ zeta potential to an average value of −21 ± 5 mV. This value correlates with the electrokinetic charge of human serum albumin, which is the most abundant human plasma protein, indicating protein adsorption independently of the initial surface charge of NP. In the current study, the zeta potential of PEGylated NT3–BDNF nanoparticles reached −10 mV ± 3 mV, which could be explained by the electrokinetic charge of proteins and growth factors, abundant in FBS, which reaches −10 mV, not −20 mV [49]. Unexpectedly, after 72 h incubation, the adsorption of negatively charged PEGylated NT3–BDNF nanoparticles at the negatively charged FBS proteins led to a greater reduction in the surface charge of nanoparticles. Those results are inconsistent with the mean-field Gouy–Chapman theory, where the adsorption process is purely driven by electrostatic interactions. However, in the case of rhNT3 and rhBDNF proteins, the adsorption process is more complex, e.g., van der Waals, electrostatic, hydrogen bond formation, and conformational changes drive the protein adsorption on like-charged surfaces [50,51]. The adsorption of negatively charged NPs on negatively charged serum proteins could be explained by highly heterogeneous charge distribution over the protein molecule, conformational changes, or chelation of cations by the PEG chains [16,52,53]. These findings are in good agreement with previous studies for nanoparticles, bearing the same zeta potential sign as the adsorbed protein [51,54].

Finally, NP functionality was determined by studying their effect on the ARPE-19 cell line. Previously, it was observed that nanoparticles cause greater cytotoxic effects in serum-free environments, as they form large aggregates, rapidly precipitate, and directly interact with cells [55]. In contrast, our study revealed that a serum-rich environment could improve the stability of NPs, resulting in prolonged cellular interactions with PEGylated NT3–BDNF complexes when compared with serum-free conditions. Therefore, using a confocal microscope, we observed increased NPs’ accumulation in cells in complex media after 72 h. It should be noted that the fluorescence is mostly localized to the cytosolic space rather than the cell plasma membrane, suggesting that nanoparticles may be internalized by cells without losing their integrity. If the nanoparticles were dissociated and released by the fluorescent probe before being taken up by the cells, the fluorescent signals would also have been observed in both the cell membrane and cytosol [56]. It is shown that ARPE-19 cells cultured in serum-free media and exposed to any concentration of PEGylated NT3–BDNF nanoparticles displayed significantly higher cytotoxicity when compared with cells in complex media. It was interesting to observe that, under the same conditions, we observed a higher level of nanoparticle aggregation. Thus, cell cytotoxicity was significantly increased after incubation with larger NPs. In contrast to our results, it was shown that gold nanoparticles bear a greater cytotoxic effect in complex media [48]. In our study, because of the use of PEG, this mechanism was altered, as PEG was shown to enhance the nanoparticles’ stability and improve neurotrophins [57]. Therefore, the PEGylated NT3–BDNF NPs were more biocompatible and stable in the cellular environment.

From the nanomedicine point of view, NPs must have abilities for gradual and controlled release of the payload. Here, the effect of serum components is of main interest owing to the formation of a protein layer on the NPs’ surfaces in vivo, which affects the nanocarrier protein release profile, cell cytotoxicity, endocytosis, and bioavailability. Previously, it was shown that protein corona formation significantly reduced drug release in the serum environment because of the protein decoration on the surface of NPs and the excess of proteins in the solution [58]. Our study confirmed that the release profile of both neurotrophins in the serum environment was more gradual when compared with the serum-free environment. Notably, the increased release of both rhBDNF and rhNT3 in vitro in the complex medium was correlated with higher cell viability of ARPE-19 cells. In contrast, in serum-free conditions, a low release profile of both NTs was associated with higher ARPE-19 cytotoxicity after 72 h. Another study, which used PGA nanoparticles, achieved a corresponding effect, showing that released rhBDNF diminished cell cytotoxicity in vitro [59,60,61].

The obtained results depicted that the PEGylated NT3–BDNF complexes introduced to the complex serum could behave similarly to pure proteins and reversibly adsorb at the macromolecules, thus participating in the process of protein corona formation and stability of the entire system. This study elucidates the potential of PEGylated NT3–BDNF nanoparticles for medical applications. However, further research is needed to evaluate their behavior in vivo and understand their interactions with the biological milieu. This research could provide valuable insights into their possible therapeutic use and sets out the way for their translation into clinical settings.

## 5. Conclusions

This study shows the effect of complex media on the physicochemical properties of novel, biodegradable PEGylated BDNF–NT3 nanoparticles in cell-free systems and the human retinal pigment epithelial cell line. We demonstrated that rapid formation of complex protein corona is time- and nanoparticle-size-dependent, consequently altering the hydrodynamic diameter and surface charge of PEGylated neurotrophin complexes.

It was found that the reversibility of NPs’ aggregation and dispersion considerably influenced their physical, chemical, and biological functionality. In serum-rich conditions, the change in the surface properties significantly decreased the aggregation of NPs.

Furthermore, neurotrophins’ release profiles in the cellular environment confirmed the gradual protein release after exposure to complex media. In contrast, serum-free conditions resulted in NPs’ clustering and instant BDNF/NT3 release in cell-free and in vitro systems. Moreover, the NP aggregates had a significantly more cytotoxic effect on ARPE-19 cells when compared with NPs in complex media.

Our method for NP formulation can be widely applied to study the adsorption of therapeutic proteins on the surface of biodegradable polymers, further enhancing protein stabilization with a low cytotoxic effect and successful cellular internalization.

## Figures and Tables

**Figure 1 pharmaceutics-15-02236-f001:**
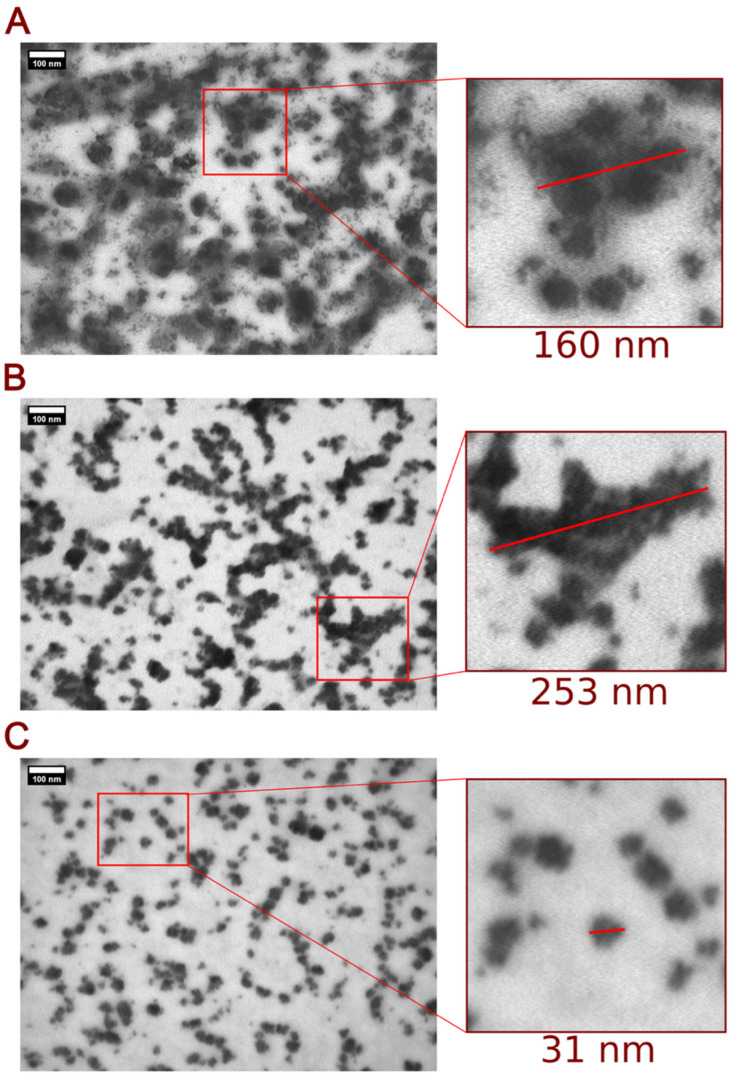
TEM images of PEGylated NT3–BDNF nanoparticles deposited from 0.15 M, pH 7.4 in PBS, containing various concentrations of recombinant neurotrophins: 0.1 mg L^−1^ NT3 and 0.1 mg L^−1^ BDNF (**C**), 1 mg L^−1^ NT3 and 1 mg L^−1^ BDNF (Appendix A), and 5 mg L^−1^ NT3 and 5 mg L^−1^ BDNF (Appendix A), and NT3 (**A**) and BDNF (**B**). The images on the left and right sides correspond to a 100,000 magnification. Scale bar is 100 nm. Note the dispersed type of PEGylated NT3–BDNF nanoparticles ((**C**), Appendix A), with aggregation of NTs (**A**,**B**).

**Figure 2 pharmaceutics-15-02236-f002:**
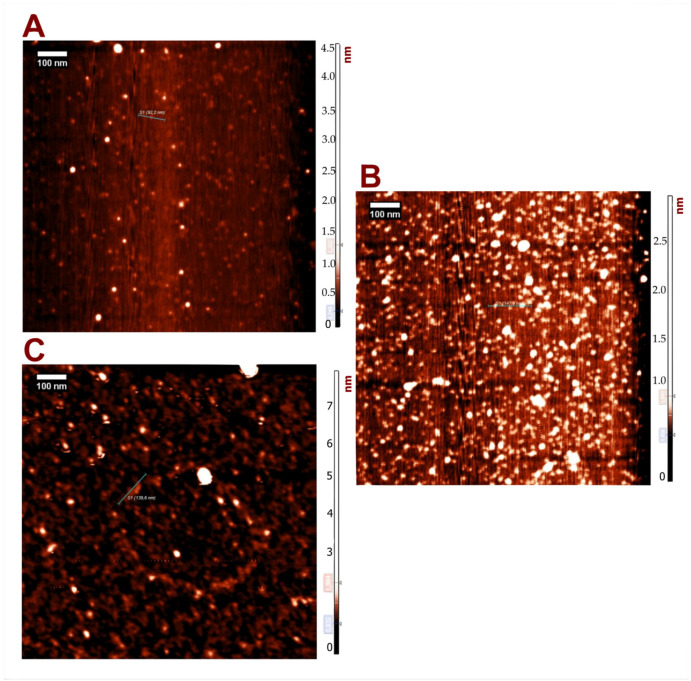
AFM images of PEGylated NT3–BDNF nanoparticles deposited onto the mica surface at 0.15 M, pH 7.4 in PBS for various concentrations of recombinant neurotrophins: 0.1 mgL^−1^ (**A**), 1 mgL^−1^ (**B**), 5 mgL^−1^ (**C**). The scale of the *x-* and *y*-axis is in the 1000 nm range. The cross-section structures of PEGylated NTs’ nanoparticles can be found in the Appendix A.

**Figure 3 pharmaceutics-15-02236-f003:**
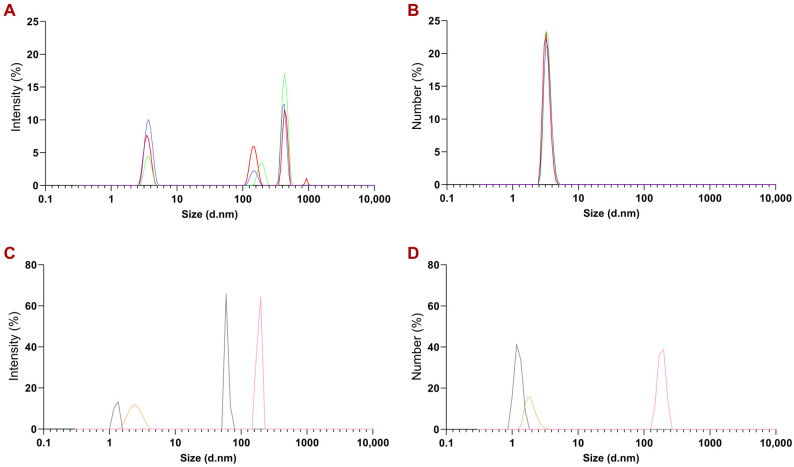
Size distribution of rhBDNF, rhNT3, PEG, and PEGylated NT3–BDNF nanoparticles by MADLS. The hydrodynamic diameter of NPs with different neurotrophin concentrations (part (**A**,**B**)): 0.1 mg L^−1^ (blue curve), 1 mg L^−1^ (green curve), and 5 mg L^−1^ (red curve). The hydrodynamic diameter of rhBDNF, rhNT3, and PEG is shown in part (**C**,**D**): rhBDNF (pink curve), rhNT3 (grey curve), and PEG (orange curve).

**Figure 4 pharmaceutics-15-02236-f004:**
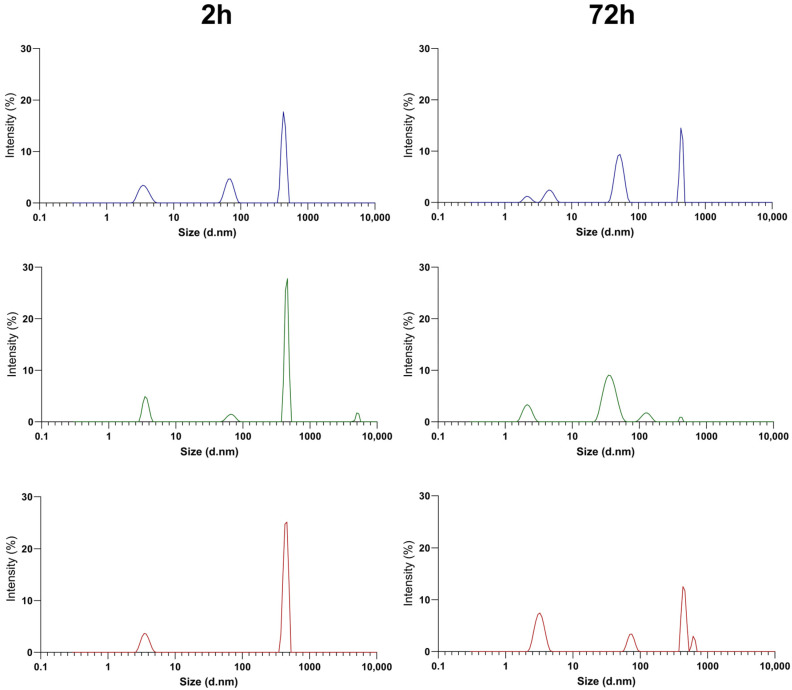
Size distribution by MADLS of PEGylated NT3–BDNF nanoparticles with different neurotrophin concentrations: 0.1 mg L^−1^ (blue curve), 1 mg L^−1^ (green curve), and 5 mg L^−1^ (red curve) over 2 h and 72 h after formulation. Each curve represents the average of the nanoparticle sizes resulting from the six syntheses. The size distribution of nanoparticles (Appendix A and Appendix A) over 24 h after formulation.

**Figure 5 pharmaceutics-15-02236-f005:**
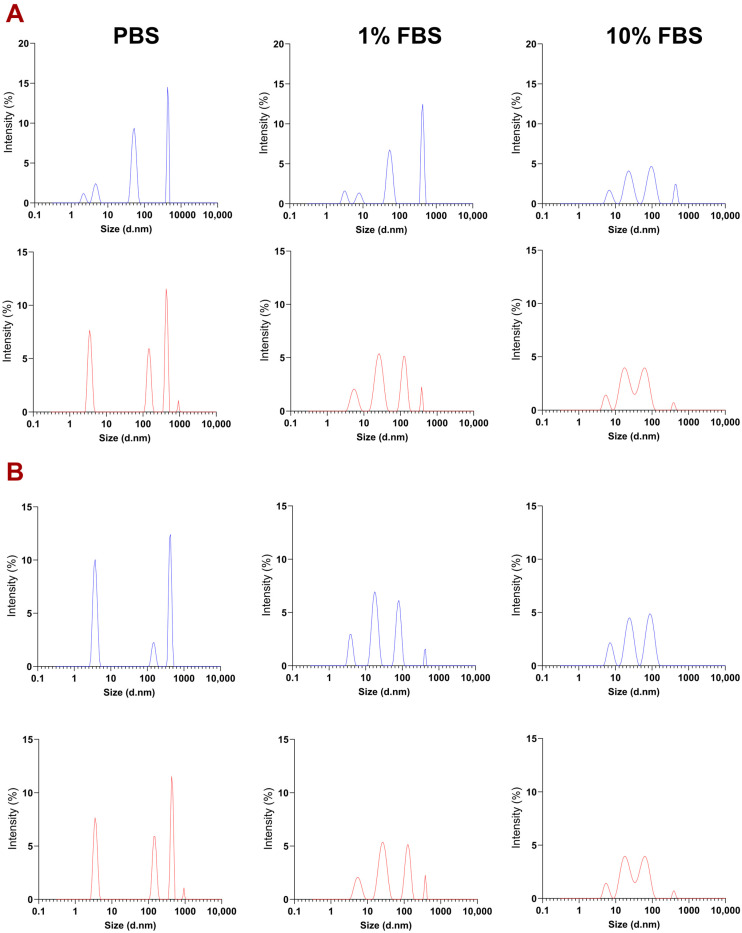
Dependence of aggregation/dispersion on various bulk conditions was shown after 30 min (part (**A**)) and 72 h (part (**B**)). PEGylated NT3–BDNF nanoparticles’ aggregation/dispersion showed variable dynamics depending on bulk condition. Neurotrophins NT3/BDNF concentrations: 0.1 mg L^−1^ (blue curve), 1 mg L^−1^ (Appendix A), and 5 mg L^−1^ (red curve). Each curve represents the average of the nanoparticle sizes resulting from the six syntheses. The concentration of PEGylated BDNF–NT3 nanoparticles as a function of NTs’ concentrations, bulk conditions, and time points (Appendix A).

**Figure 6 pharmaceutics-15-02236-f006:**
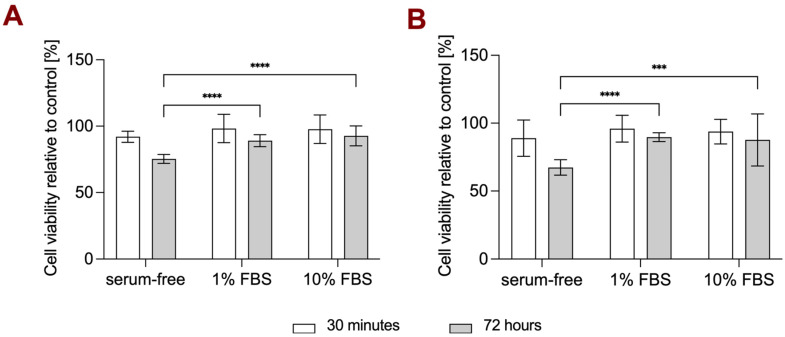
Effect of serum concentration and PEGylated NT3–BDNF nanoparticles on ARPE-19 cell viability over 72 h with the following concentrations of NTs: (**A**) 0.1 mg L^−1^ and (**B**) 5 mg L^−1^ (*n* = 12). Statistical analysis: two-Way ANOVA with post-hoc Tukey’s test (in grey) for time-course and (in white) for serum concentration analysis: **** *p* < 0.0001 and *** *p* < 0.001.

**Figure 7 pharmaceutics-15-02236-f007:**
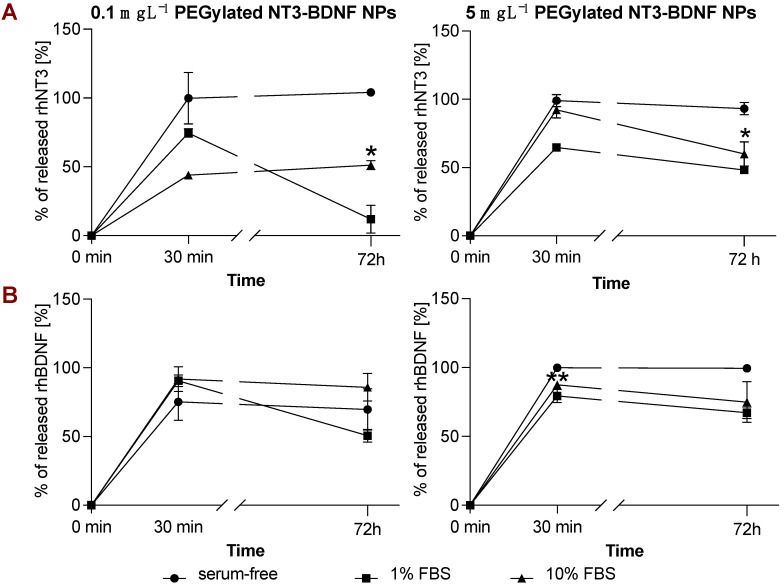
Release of rhNT3 (**A**) and rhBDNF (**B**) from PEGylated NT3–BDNF nanoparticles in a cell-free system. PEGylated NT3–BDNF nanoparticles (0.1 mg L^−1^ and 5 mg L^−1^) were incubated in serum-free, 1% FBS, and 10% FBS media for up to 72 h with constant mixing. Data were analyzed using repeated measures ANOVA with Tukey’s post-test comparing protein release profile in media supplemented with 1% or 10% FBS to serum-free media; * *p* < 0.05 and ** *p* < 0.01.

**Figure 8 pharmaceutics-15-02236-f008:**
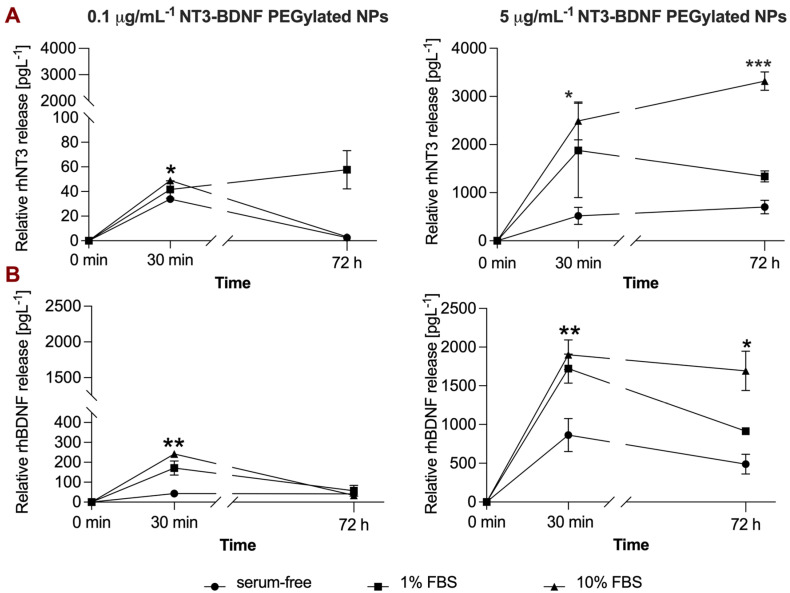
PEGylated NT3–BDNF NP release profile in vitro. NPs (0.1 µgmL^−1^ and 5 µgmL^−1^) show time-dependent release over a 72 h period for rhNT3 (**A**) and rhBDNF (**B**). The results are mean ± SD of three replicates. Data were analyzed using repeated measures ANOVA with Tukey’s post-test; * *p* < 0.05, ** *p* < 0.01, and *** *p* < 0.001.

**Figure 9 pharmaceutics-15-02236-f009:**
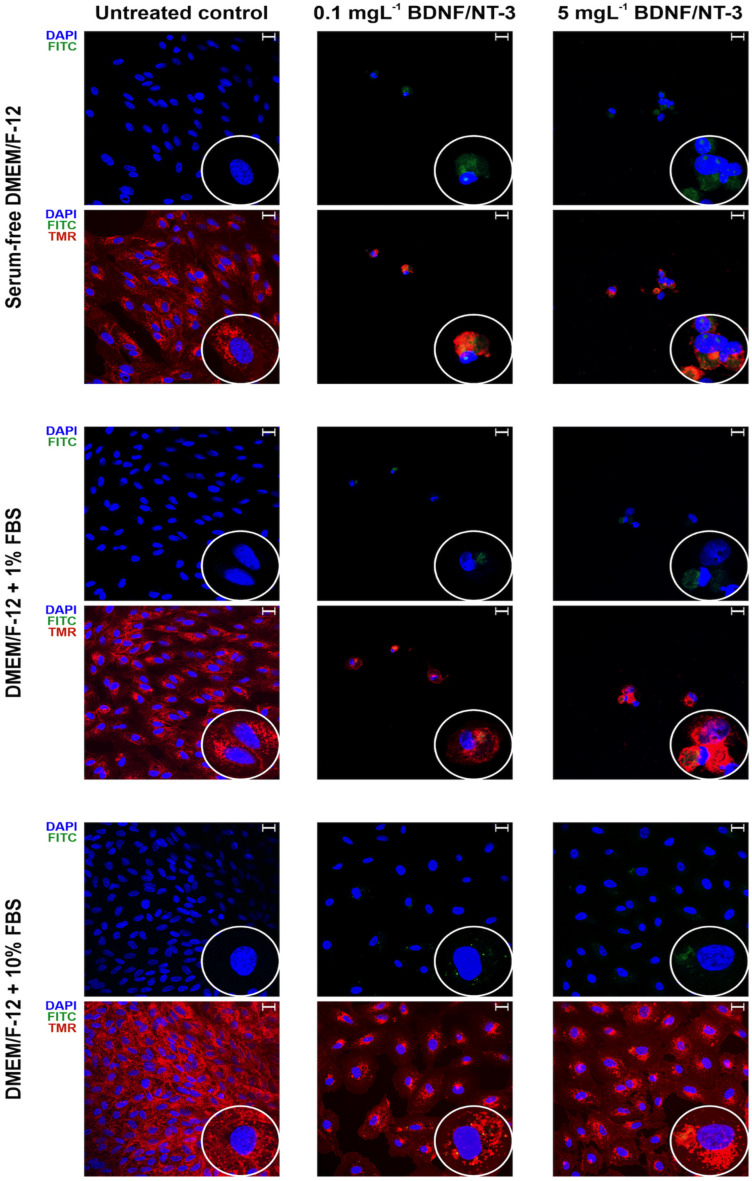
Cellular internalization of PEGylated BDNF/NT3 nanoparticles after 72 h. The ARPE-19 cells cultured in different DMEM/F-12 media conditions were incubated for 72 h with PEGylated NT3–BDNF–FITC nanoparticles. Green fluorescence (FITC) represents nanoparticles, blue (DAPI)—cell nuclei, red (tetramethylrhodamine)—surface glycoproteins. The scale bar is 20 μm. All images include inserts with 3× magnification.

**Table 1 pharmaceutics-15-02236-t001:** The correlations between zeta potential (**ζ**) of PEGylated NT3–BDNF nanoparticles as a function of various neurotrophin concentrations, bulk conditions (PBS, PBS supplemented with 1% FBS, and PBS supplemented with 10% FBS), and time.

ζ [mV]	2 h			24 h			72 h		
	PBS	1% FBS	10% FBS	PBS	1% FBS	10% FBS	PBS	1% FBS	10% FBS
PEGylated NT3/BDNF (0.1 mg L^−1^)	−8.7 ± 2.6	−11.8 ± 4.30	−5.99 ± 1.33	−10.8 ± 2.52	−7.6 ± 1.3	−9.7 ± 2.2	−5.4 ± 1.65	−9.6 ± 2.1	−9.7 ± 2.2
PEGylated NT3/BDNF (1 mg L^−1^)	−14.1 ± 1.52	−12.1 ± 2.7	−6.4 ± 3.7	−11.8 ± 3.1	−12.3 ± 3.7	−10.1 ± 2.4	−3.36 ± 1.40	−7.16 ± 2.51	−8.42 ± 2.12
PEGylated NT3/BDNF (5 mg L^−1^)	−13.9 ± 1.8	−9.82 ± 3.9	−7.1 ± 3.9	−9.8 ± 0.5	−10.2 ± 3.9	−11.7 ± 1.06	1.9 ± 3.1	−8.96 ± 1.27	−11.46 ± 1.84

## Data Availability

All data needed to evaluate the conclusions in the paper are presented in the paper.

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
