# Peer review of "The Impact of Serum Protein Adsorption on PEGylated NT3–BDNF Nanoparticles—Distribution, Protein Release, and Cytotoxicity in a Human Retinal Pigmented Epithelial Cell Model"

_pharmaceutics, 2023, doi:10.3390/pharmaceutics15092236_

Round 1

Reviewer 1 Report (New Reviewer)

The manuscript by DÄ…bkowska et al. reported the interactions between NT3, BDNF, and PEG in a cell-free system and a retinal pigmented epithelium cell line. It provides some insights into the role of protein corona with implications for cytotoxicity, NP’s distribution, and release profiles of the PEGylated nano-bio interface. In general, the work is worth doing; however, the quality of the manuscript needs substantial improvements. Some more specific comments are below:

1) The formation mechanism of the nanoparticles should be discussed.

2) Improvements to scholarly writing are required throughout the manuscript. For instance, “a 0.22 mM filter.”, should be a 0.22 µm filter; “mgL-1”, a space or a / between the units is required; “at the concentration of 10,000 125 mgL-1”, what does the concentration mean? A mass concentration is suggested.

3) How is the SEM sample prepared? As the nanoparticles were suspended in 0.15 M PBS, the PBS salt may interfere with the particles after drying

4)  Most of the figure captions need rewriting; they are very hard to read and thereafter too difficult to understand the data illustrated.

5) Why does the rhBDNF molecule have such a large size of 301.7 nm?

6) How many repeated measurements were performed for the samples shown in Figures 3-5? Many of the data points look quite odd.

7) In figures 7 and 8, why is the release amount over 72 hours less than that for 30 minutes in some samples?

Improvements to scholarly writing are required throughout the manuscript. For instance, “a 0.22 mM filter.”, should be a 0.22 µm filter; “mgL-1”, a space or a / between the units is required; “at the concentration of 10,000 125 mgL-1”, what does the concentration mean? A mass concentration is suggested.

Author Response

Please, find enclosed our revised paper entitled “The Impact of Serum Protein Adsorption on PEGylated BDNF-NT3 Nanoparticles - Distribution, Protein Release, and Cytotoxicity in Human Retinal Pigmented Epithelial Cell Model ” together with the response to the Reviewer’s comments.

Reviewer 2 Report (New Reviewer)

The presented article will certainly be of interest to the readers of the journal.

The positive features of the manuscript include a high level of reliability and a wide array of experimental data obtained by the authors.

To improve the manuscript, I propose the following minor improvements:

1) In the introduction, I would like to add a more detailed description of the opsonization process

2) There are design flaws in the text: font color and size, typos

3) The authors mention that earlier data were obtained on the independence of the protein corona from time, while the opposite results were obtained in the work. I would like to see the position of the authors explaining this difference in more detail.

Author Response

Please, find enclosed our revised paper entitled “The Impact of Serum Protein Adsorption on PEGylated BDNF-NT3 Nanoparticles - Distribution, Protein Release, and Cytotoxicity in Human Retinal Pigmented Epithelial Cell Model ” together with the response for the Reviewer’s comments.

Reviewer 3 Report (New Reviewer)

I believe it is a valuable manuscript reporting carefully designed in vitro examinations. However, certain points of the manuscript need improvements:

** The title suggests that the topic of the manuscript is about protein corona. To the field of nanomedicine, protein corona mostly refers to a layer of protein with diverse compositions formed around nanoparticles upon exposure to physiologically meaningful environments, such as serum. While FBS is used throughout the manuscript for such investigations, there is no characterization of protein corona formation – so I feel the title is somewhat misleading. I recommend revising it to make it fit the content of manuscript better, such as replacing it with “protein adsorption”.

** SEM and TEM are two entirely different characterization modalities, with distinct mechanisms. The instrument used by the authors may be compatible for both modalities, but the images shown in Figure 1 are TEM images – please revise accordingly.

** It is unclear why the cell membrane stain (TMR) was included in Figure. 9. Please elaborate more on the rationales of including this. What information does this staining provide?

** The interpretation of the zeta-potential data shown in Table 1 may not be accurate. Zeta-potential values around 0, no matter positive or negative, should be treated as near neutral because the entire nanoparticle population holds a distribution of zeta-potential, and only relative comparison is meaningful. The authors should refer to this report (10.1016/j.jconrel.2016.06.017) about this point.

Author Response

Please, find enclosed our revised paper entitled “The Impact of Serum Protein Adsorption on PEGylated BDNF-NT3 Nanoparticles - Distribution, Protein Release, and Cytotoxicity in Human Retinal Pigmented Epithelial Cell Model ” together with the response to the Reviewer’s comments

Round 2

Reviewer 1 Report (New Reviewer)

Comments for this reviewer have been basically addressed.

Comments for this reviewer have been basically addressed.

Reviewer 3 Report (New Reviewer)

The authors have addressed my points raised in the first round of review. The manuscript can now be accepted for publication.

This manuscript is a resubmission of an earlier submission. The following is a list of the peer review reports and author responses from that submission.

Round 1

Reviewer 1 Report

The Impact of Protein Corona Formation on PEGylated BDNF-NT3 Nanoparticles Distribution, Therapeutic Protein Release and Cytotoxicity in Human Retinal Pigmented Epithelial Cell Model 

Summary: The authors describe formulations of neurotrophins BDNF and NT3 containing PEG. The introduction is written well. However, the manuscript lacks clear goals, experimental design is poor, data presentation requires improvement and manuscript is poorly written. Hence, is not fit for publication in its current state.

Major comments:

1)    The manuscript lacks goals, scope and potential impact. Hence the choice the experiments (studying effects of protein concentrations in the formulations for example) is confusing

2)    With respect to nanoparticle synthesis, there is lack of evidence for PEG adsorption. How is PEG adsorption on proteins being confirmed? There is also lack of evidence for BDNF and NT3 co-assembly. Additionally, are the particles always suspended in the 1% w/v PEG solution even after “adsorption” process? This is not clear.

3)    Fig 1 lacks proper negative controls. Recombinant proteins (BDNF and NT3) routinely have aggregates. Hence, it is essential to prove they are not the source of particles seen here. Additionally, PEG is known to induce protein precipitation. It is essential to prove the particles in Fig 1 are nanoparticles and not aggregates. 

4)    The manuscript lacks evidence that the particles seen are biologically active. Hence, these could be inactive protein aggregates

5)    Fig 1 and 2: poorly presented. Please refer to published articles for reference

6)    Fig 3 and 4: quality too poor to read, poor presentation. Presence of large particles in DLS further confirms aggregation in the formulations

7)    Fig 6 and 7: When measuring “released” BDNF and NT3 in media, how is free vs. particle bound protein being differentiated? These experiments require additional scrutiny

8)    Method for FITC labeling of particles involves NHS-PEG-FITC. Hence, NHS mediated covalent coupling of proteins-PEF occurs in solution. Thereby, particles labeled by this method cannot be representative.

9)    Fig 8, 9, 10: There is no signal in FITC channel. Hence, these figures can be removed as they do not provide any information on the particles. Experimental conditions need to be optimized

10) Results section throughout the manuscript should be re-written by focusing on main conclusions from data and how they support the goals of the paper. All figures need to be redone for better quality and easier interpretation

Reviewer 2 Report

Dear authors,

The manuscript you proposed describes the impact of the formation of the corona protein on PEGylated BDNF-NT3 Nanoparticles. The research topic is very interesting and the methods and techniques used are correct. The chemical identity of nanoparticles changes when they come into contact with a biological fluid, in this case with FBS. Understanding how the behavior of cells changes in the presence of soft corona and hard corona on nanoparticles is one of the fundamental points of nanomedicine.

Comments:

1. Reading the article was not clear and in some places I had to write with pen and paper in order to understand the data well. Authors should include less relevant data in the supporting information to generate less confusion.

2. In figure 1 the bar scales are not shown and the incubation time is not indicated in the caption.

3. Check the brand of the SEM is JEOL and not AJEOL

4. Figure 2 improving the graphics of the AFM image structure simply show the 2D figure and show the section where you have extracted the profile. Profile graphics are not legible.

5. The AFM and SEM images shown are only PEGylated BDNF-NT3 Nanoparticles with the different concentrations only in PBS, why didn't the authors also analyze the PEGylated BDNF-NT3 Nanoparticles incubated in 1% FBS and 10% FBS? The authors should delete the 3D image of the AFM and add the images of the PEGylated BDNF-NT3 Nanoparticles with the different concentrations in FBS.

6. In Figures 3 and 4 the graphs are not readable.

7. In the paragraph of the results on page 10 for all the data that are discussed I suggest summarizing them in a table, to help reading the article, or to write the data more clearly.

8. In table 4, the standard deviation must be placed next to the value not in another row, furthermore, the table is not recalled in the text.

9. Page 18 line 672 check the range of values.

10. Figures 8,9 and 10 are not referred to in paragraph 3.5.

11. The images in figures 8,9 and 10 do not have the right magnification to observe the FITC labeled fluorescence BDNF-NT-3 - PEG. Authors must insert high-magnification images.

12. Be careful in formatting superscripts and subscripts in the manuscript. For example line 104,107, 119, 121 ...

13. Why did the authors not analyze the different formation of the protein corona in the three cases (serum free, 1% FBS and 10FBS%) at 72h of incubation, by SDS-PAGE? Knowing which proteins form the corona protein on PEGylated BDNF-NT3 Nanoparticles helps to better understand the interaction of cells with them and consequently to have stronger conclusions.

Reviewer 3 Report

The manuscript entitled "The Impact of Protein Corona Formation on PEGylated BDNF-NT3 Nanoparticles Distribution, Therapeutic Protein  Release and Cytotoxicity in Human Retinal Pigmented Epithelial Cell Model" is very interesting and well designed formulation approach with mechanistic proof. The authors address some minor revisions.

1. Write the materials section in the manuscript.

2. The abbreviations used was not uniform. Make sure.

3. Write the reason for selection of NPs for the study not discussed.

4. Write the method of distribution used for the PS measurement through Zeta sizer.

5. In statistical analysis, the data subjected to p less than or equal to. Rewrite in the methods section.

6. Figure 1 - the image resolution and scale bar unable to read. Provide the figure with more clarity.

7. Figure 3 not necessary for the study.

Round 2

Reviewer 1 Report

Thank You to the authors for attempting to improve the manuscript. However, the revised manuscript is not fit for publication. Major comments made earlier were not addressed adequately: 1) No evidence was provided to support co-assembly of BDNF and NT3, which is critical to the manuscript 2) no improvement in quality of figures 3) unsatisfactory response to comments 7-9
